# A novel form of JARID2 is required for differentiation in lineage-committed cells

Diaa Al-Raawi[1], Rhian Jones[1], Susanne Wijesinghe[1], John Halsall[2], Marija Petric[1], Sally Roberts[2], Neil A Hotchin[1] iD & Aditi Kanhere[1,*] iD

## Abstract

Polycomb repressive complex-2 (PRC2) is a group of proteins that play an important role during development and in cell differentiation. PRC2 is a histone-modifying complex that catalyses methylation of lysine 27 of histone H3 (H3K27me3) at differentiation genes leading to their transcriptional repression. JARID2 is a co-factor of PRC2 and is important for targeting PRC2 to chromatin. Here, we show that, unlike in embryonic stem cells, in lineage-committed human cells, including human epidermal keratinocytes, JARID2 predominantly exists as a novel low molecular weight form, which lacks the N-terminal PRC2-interacting domain (ΔN-JARID2). We show that ΔN-JARID2 is a cleaved product of full-length JARID2 spanning the C-terminal conserved jumonji domains. JARID2 knockout in keratinocytes results in up-regulation of cell cycle genes and repression of many epidermal differentiation genes. Surprisingly, repression of epidermal differentiation genes in JARID2-null keratinocytes can be rescued by expression of ΔN-JARID2 suggesting that, in contrast to PRC2, ΔN-JARID2 promotes activation of differentiation genes. We propose that a switch from expression of full-length JARID2 to ΔN-JARID2 is important for the up-regulation differentiation genes.

**Keywords** cell differentiation; JARID2; N-terminal domain; polycomb; proteolytic cleavage
**Subject Categories** Chromatin, Epigenetics, Genomics & Functional Genomics; Stem Cells
The EMBO Journal (2019) 38: e98449

## Introduction

Polycomb group (PcG) proteins are very important transcriptional repressors and play a crucial role in regulating gene expression during development (Margueron & Reinberg, 2011; Holoch & Margueron, 2017). They function by catalysing histone modifications that result in repressive chromatin and down-regulation of neighbouring genes. Polycomb group proteins form two major

complexes, polycomb repressive complex-1 (PRC1) and polycomb repressive complex-2 (PRC2). PRC2 functions by catalysing trimethylation of histone H3 at lysine 27 (H3K27me3; Simon & Kingston, 2013). Polycomb repressive complex-2 consists of four core proteins, SUZ12, EED, RbAp46/48 and the catalytic subunit EZH2. At the molecular level, how PRC2 is recruited to its sites of action is not yet completely clear. Recent proteomic studies have revealed that PRC2 transiently associates with many proteins such as MTF2, EPOP, AEBP2 and JARID2 that typically interact with PRC2 in a mutually exclusive fashion resulting in different subclasses of PRC2 (Kim et al, 2009; Peng et al, 2009; Shen et al, 2009; Landeira et al, 2010; Li et al, 2010; Pasini et al, 2010; Walker et al, 2010; Casanova et al, 2011; Landeira & Fisher, 2011; Beringer et al, 2016; Grijzenhout et al, 2016; Liefke et al, 2016; Holoch & Margueron, 2017). Although the molecular roles of many of these interacting proteins are not well understood, many of them can modulate enzymatic activity or recruitment of PRC2 to chromatin (Holoch & Margueron, 2017).

JARID2 is required for recruitment of PRC2 to chromatin in embryonic stem cells (Peng et al, 2009; Shen et al, 2009; Landeira et al, 2010; Li et al, 2010; Pasini et al, 2010; Landeira & Fisher, 2011; Holoch & Margueron, 2017). Multiple studies in mouse and human show that N-terminal region of JARID2 (Fig 1A) is required for PRC2 recruitment and modulation of PRC2 activity (Cooper, Son et al, 2013; Kaneko et al, 2014a; da Rocha et al, 2014; Sanulli et al, 2015; Grijzenhout et al, 2016). The N-terminal region consists of a nucleosomal binding domain and a RNA binding domain that together modulate PRC2 binding to genomic DNA (Son et al, 2013; Kaneko et al, 2014a; da Rocha et al, 2014). In addition, recently it has been shown that this region of JARID2 is needed for recruitment of PRC2- to PRC1-modified nucleosomes (Cooper et al, 2016). It is clear from multiple studies that removal of JARID2 results in reduced occupancy of PRC2 on chromatin (Peng et al, 2009; Shen et al, 2009; Landeira et al, 2010; Li et al, 2010; Pasini et al, 2010; Landeira & Fisher, 2011). But surprisingly, JARID2 removal does not result in significant and consistent changes in H3K27me3 levels (Landeira & Fisher, 2011). Although in some studies JARID2 depletion in ES cells is observed to decrease H3K27me3 levels (Landeira et al, 2010; Pasini et al, 2010), other studies have reported either no change (Peng et al, 2009) or increased levels of H3K27me3 upon

1 School of Biosciences, University of Birmingham, Birmingham, UK
2 Institute of Cancer and Genomic Sciences, University of Birmingham, Birmingham, UK
 *Corresponding author. Tel: +44 121 4145896; E-mail: a.kanhere@bham.ac.uk

JARID2 removal (Peng *et al*, 2009; Shen *et al*, 2009). Further adding to this lack of clarity on the role of JARID2 in modulation of PRC2 activity, in *in vitro* studies JARID2 appears to inhibit (Peng *et al*, 2009; Shen *et al*, 2009) as well as activate (Li *et al*, 2010) the methyltransferase activity of EZH2. It has been suggested that JARID2's N-terminal domain interacts with RNAs as well as nucleosomes (Son *et al*, 2013; Kaneko *et al*, 2014b) and its post-translational modifications determine its effect on PRC2 activity (Sanulli *et al*, 2015). A recent study has also shown that, in mouse ES cells, JARID2 can modulate PRC2 activity through its interaction with another histone methylase, setDB1 (Fei *et al*, 2015). JARID2-setDB1 interaction has also been identified in lineage-committed cells including lymphocytes (Macian *et al*, 2002; Pereira *et al*, 2014) and cardiomyocytes (Mysliwiec *et al*, 2011) where JARID2 is shown to modulate other histone modifications such as H3K9me3.

The C-terminal of JARID2 has three conserved domains (jmjN, ARID, jmjC) that are characteristic of the jumonji family of histone modifiers (Fig 1A), which catalyse demethylation of histones. The C-terminal ARID domain of JARID2 is required for DNA binding. In addition, JARID2 C-terminus also has a zinc finger domain, which is needed for its interaction with SUZ12, another component of PRC2 (Peng *et al*, 2009). The jmjC domain is required for demethylase activity in other jumonji family members. However, two amino acid changes in JARID2's demethylase domain are thought to render it inactive (Klose *et al*, 2006; Landeira & Fisher, 2011).

Despite its lack of demethylase activity, JARID2 acts as an important regulator of gene expression in embryonic stem (ES) cells where it is needed for cell signalling networks necessary for maintaining the pluripotent state (Sun *et al*, 2008; Assou *et al*, 2009; Yaqubi *et al*, 2015; Sahu & Mallick, 2016). Consistent with this, a recent report suggests that forced expression of JARID2 alongside PRDM14, ESRRB and SALL4A can efficiently induce pluripotency in fibroblasts (Iseki *et al*, 2016). More importantly, a number of publications have shown that JARID2-deleted ES cells either cannot differentiate or are delayed in differentiation (Peng *et al*, 2009; Shen *et al*, 2009; Landeira *et al*, 2015; Sanulli *et al*, 2015). These findings reflect a crucial role of JARID2 in early embryonic development at the onset of ES cell differentiation. Indeed, JARID2 is indispensable for normal embryonic development and its deficiency leads to deformation of several tissues in mice as well as in humans (Jung *et al*, 2005; Takeuchi *et al*, 2006; Landeira & Fisher, 2011). Embryos with a complete loss of JARID2 either do not survive or die soon after the birth (Jung *et al*, 2005; Takeuchi *et al*, 2006; Shen *et al*, 2009; Landeira *et al*, 2010).

Although the importance of JARID2 in ES cells has been established, its role in lineage-committed cells has not been well studied, mainly because of its much lower level of expression or perceived absence (Zhang *et al*, 2011; Son *et al*, 2013). In this study, we show that, in many lineage-committed cells including human epidermal keratinocytes, JARID2 predominantly exists as a low molecular weight (LMW) form. In the LMW form, the N-terminal region is cleaved from full-length JARID2 resulting in a stable C-terminal fragment (ΔN-JARID2). This form of JARID2 lacks N-terminal nucleosomal and RNA binding domains (Son *et al*, 2013; Kaneko *et al*, 2014a), implying a substantial effect on JARID2 functionality and its interactions with PRC2 complex. Consistent with this, a recent study showed that C-terminal region of JARID2 cannot restore H3K27me3 marks (Cooper *et al*, 2016). We show that the level of ΔN-JARID2

increases as differentiation of keratinocytes progresses. We find that JARID2 knockout results in impaired differentiation of keratinocytes and this effect is reversed by expression of ΔN-JARID2, indicating that this form of JARID2 is needed for activation of polycomb target genes during differentiation.

# Results

## A low molecular weight form of JARID2 exists in lineage-committed cells

JARID2 has been extensively studied in embryonic stem (ES) cells where it is reported as a 140 kDa protein (Fig 1B). It is thought that JARID2 is not expressed or is expressed at very low levels in lineage-committed cells (Zhang *et al*, 2011; Son *et al*, 2013). Therefore, we investigated JARID2 mRNA as well as protein expression in multiple types of lineage-committed human cells (Fig 1B and C). In most cell types, JARID2 mRNA is present at detectable but lower levels than in the ES cells (Fig 1C). Surprisingly, when we investigated protein levels in lineage-committed cells (Fig 1B), we detected another band at around ~80 kDa that has not been reported previously. We observed that in the majority of cell types we studied, this band is much more dominant than the 140 kDa band corresponding to the canonical full-length JARID2 isoform-1 (Fig 1B). To verify that this low molecular weight form is encoded by JARID2 and is not a non-specific band cross-reacting with our antibody, we transfected keratinocytes (HaCaT cells), HEK293T and K562 cells (Figs 1D and EV1A and B) with different JARID2 siRNAs designed to target the 5′ (exon 3) and 3′-ends (exon 15) of JARID2 mRNA (Appendix Table S1). Transfection of both siRNAs resulted in disappearance of the ~80kDa band along with the canonical 140 kDa band (Figs 1D and EV1A and B). The siRNA-mediated knockdown and Western blot with an alternative JARID2 antibody (Fig EV1B) confirmed that this is a low molecular weight (LMW) form of JARID2. To rule out the possibility that this band is a degradation product of JARID2, we also extracted protein in the presence of increasing amount of protease inhibitor and observed no difference in the levels of the LMW form (Fig EV1C).

## LMW JARID2 is not a transcriptional isoform

Next, we sought to understand whether the LMW form is translated from a distinct transcript of JARID2. According to latest ENSEMBL annotations of human genes, three different mRNA isoforms of JARID2 have been predicted (Fig 2A; Rosenbloom *et al*, 2015). According to size predictions, the three isoforms would produce proteins of sizes 140, 120 and 106 kDa, much larger than ~80 kDa indicating that this LMW form might not correspond to one of the annotated isoforms of JARID2. In addition, the mRNA for isoform-3 does not express exon 15, which is targeted by one of the siRNAs used in our knockdown experiment (Figs 1D and 2A), indicating that the ~80 kDa form is not a product of isoform-3. To test whether it might be a product of a mRNA variant transcribed from an internal promoter that is not yet part of current annotations, we analysed a large collection of CAGE (cap analysis gene expression) tag data that is available through the ENCODE database (Rosenbloom *et al*, 2013). In CAGE analysis, short fragments from 5′ ends of capped

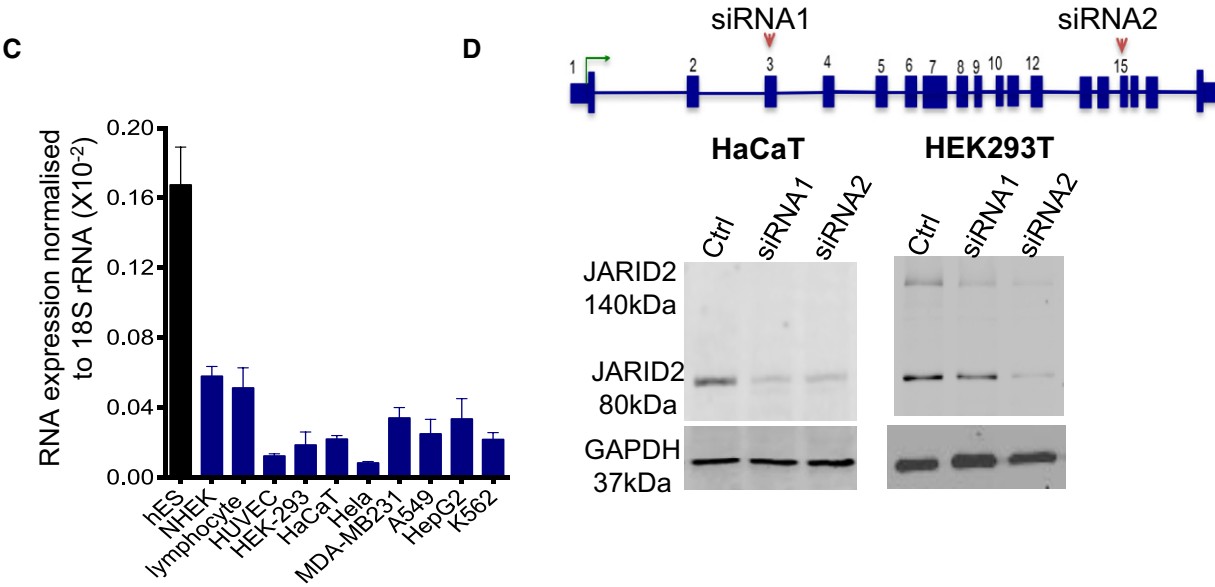

**Figure 1.   JARID2 exists as a LMW form (~80 kDa) in many cell types.**

A   A schematic diagram showing functional domains of JARID2: UIM, ubiquitin interaction motif; transcription repressive domain consisting of sub-domains 1-3; JmjN, JumonjiN; ARID, AT-rich DNA binding domain; GSGFP which is a SUZ12 binding domain; JmjC, JumonjiC; and ZF, zinc finger domain. Domains 1-3 constitute PRC2-associated domains. Domain 1 is required for PRC2 stimulation, domain 2 is needed for binding to EZH2, and domain 3 is needed for nucleosomal binding. The RNA binding domain has been indicated with dashed box.

B   Immunoblot of whole-cell lysates from multiple human lineage-committed cell lines using a C-terminal anti-JARID2 antibody. The size of canonical form is 140kDa, but an additional strong band was detected at around ~80 kDa. The blot is divided according to cell types, which are indicated in the figure.

C   qPCR measurements of JARID2 mRNA levels in different cell lines. The RNA levels were measured relative to 18S rRNA. Means and SEs are calculated over at least two independent experiments and three technical replicates. The data are represented as mean $\pm$ SE.

D   Immunoblot showing that the ~80 kDa band disappears upon treatment of HEK293T and HaCaT cells with siRNAs against exon 3 and exon 15 of JARID2 RNA. Non-silencing control (NSC) was used as a transfection control. Densitometric measurements corresponding to this experiment are shown in Fig EV1A.

Source data are available online for this figure.

RNAs are sequenced and mapped back to the genome to find transcription start sites. CAGE tag data clearly indicate that in most of the cell types we tested (Fig 1B), JARID2 is predominantly transcribed from single transcription start site matching that of JARID2 isoform-1 (Fig 2B). We also designed primers to specifically amplify each isoform of JARID2 (Appendix Table S2) and carried out qPCR analysis to detect levels of the three different isoforms. Our qPCR measurements further confirmed that isoform-1 is the predominant isoform in these cells (Fig 2C). To rule out the possibility that the ~80 kDA band is a product of an mRNA with same transcription start site as isoform-1, but with a different splicing pattern, we amplified JARID2 mRNA using RT–PCR with primers in exon 3 and exon 15 (Appendix Table S3) which, according to our knockdown experiments, are part of mRNA that produces this low molecular weight isoform (Fig 1D). Using these primers, only one product was amplified and its size as well as sequence confirmed that it contained all the exons between exon 3 and exon 15 (Fig 2D, see Appendix). In addition, RT–PCR using primers in the first and the last exon also amplified only one product of the expected size (Fig 2D). This confirmed that the LMW form of JARID2 is a product of mRNA with a complete set of exons as in the case of full-length mRNA of isoform-1. This suggested that LMW JARID2 is either a cleaved protein product of full-length JARID2 or is translated from an internal site that is distinct from the isoform-1 translation site.

## LMW JARID2 is a cleaved product of isoform-1

To check whether JARID2 mRNA is translated from two distinct translation start sites, one corresponding to the reported 140 kDa product and other corresponding to the ~80 kDa low molecular weight form, we analysed published ribosome profiling data (Michel *et al*, 2015). We could only identify a single ribosome initiation site, which corresponded to the 140 kDa product (Fig 2E). Taking into consideration the above observations and the fact that we used a JARID2 antibody that recognises the C-terminal region of JARID2, we can predict that a ~80 kDa protein can be produced from translational start site which is far downstream of translational start site of isoform-1. However, if LMW JARID2 is a cleaved product of

full-length JARID2, rather than a distinct isoform translated from an internal start site, mutating the translation start site of JARID2 isoform-1 should also knockdown LMW JARID2. To test this, we knocked out isoform-1 by CRISPR/Cas9-mediated targeting of its translation start site (Fig EV2A and B). In knockout (KO) cells, where we targeted the translation start site of isoform-1, we detected the same level of mRNA as wild type (Fig EV2C). However, the LMW JARID2 protein band disappeared (Figs 2F and EV2D). This supports the hypothesis that the 80 kDa LMW form and JARID2 isoform-1 have identical translation start sites and that the LMW form is a cleavage product of JARID2 isoform-1.

To further verify that the LMW form is indeed a cleavage product of JARID2 isoform-1, we transfected HaCaT cells with N-terminally flag-tagged ORF of full-length JARID2 (FL-JARID2) isoform-1 expressing plasmid vector. If LMW JARID2 is truly derived from full-length JARID2, we predicted that exogenous expression of full-length JARID2 would also result in an increase in the levels of the LMW form. Consistent with this hypothesis, we observed that LMW levels increased 2- to 3-fold upon exogenous expression of full-length JARID2 (Figs 2G and EV2E and F). However, in a blot with anti-Flag antibody, which should detect the N-terminal tag on JARID2, we could not detect the full-length JARID2, 80 kDa band or any other low molecular weight product (Fig EV2G), indicating that the N-terminal portion of JARID2 is missing in these cells. To further confirm this, we carried out mass spectrometry identification of the LMW band. In this experiment, we detected peptides spanning only the C-terminal of JARID2 (Figs 2H and EV3). All these observations strongly support the hypothesis that the LMW form is a cleaved product of full-length JARID2 isoform-1 and is missing the N-terminal portion. Accordingly, we designated the LMW form ΔN-JARID2. To confirm the size of ΔN-JARID2 and to estimate the cleavage position, we prepared plasmid constructs expressing different length C-terminal fragments of JARID2 (Appendix Fig S1A) and checked if their predicted protein products (103, 88 and 79 kDa) run at a similar size to ΔN-JARID2 (Appendix Fig S1B). We found that the fragment spanning 554-1,246 amino acids produced a protein product (79 kDa) that co-migrated with ΔN-JARID2 confirming that the size of ΔN-JARID2 is ~80 kDa. This is also consistent with the mass

---

**Figure 2. JARID2 LMW is a cleaved product of full-length JARID2 isoform-1.**

A   A schematic showing three different isoforms of human JARID2, as predicted by ensembl annotations. The translation start site of each isoform is labelled using green arrow. Predicted sizes (140, 120 and 106 kDa) of proteins corresponding to three isoforms have been indicated.

B   CAGE-seq data showing the transcription start sites of JARID2 in different cell lines. CAGE peak is mainly observed at JARID2 isoform-1 in most cell types. JARID2 gene structure is shown at the top.

C   qPCR measurements of RNA levels (*n* = 3) for the three isoforms of JARID2 in HaCaT cells. The RNA levels are plotted relative to 18S rRNA. Level of JARID2 isoform-1 is significantly higher than that of isoform-2 (\*\*\*$P < 0.001$) and isoform-3 (\*\*\*\*$P < 0.0001$). Data for three independent experiments are represented as mean ± SE. Multiple comparisons were performed and *P*-values were calculated using one-way ANOVA.

D   Agarose gel showing RT–PCR products amplified using primers in exon 3 and exon 15 (2.9 kb) as well as RT–PCR product corresponding to JARID2 isoform-1 (3.7 kb) amplified using the first (exon 1) and the last exon (exon 18). Only one product at the right size is observed indicating that it is unlikely that JARID2 might be a product of an alternatively spliced isoform of JARID2.

E   Ribosomal profiling data mapped on JARID2 isoform-1 and isoform-2. Main translation site corresponding to isoform-1 is highlighted using a dashed box.

F   CRISPR-Cas9 knockout of JARID2 using a sgRNA guide designed to target main translation start site as seen in (E). Immunoblot revealed that ~80 kDa form was removed from JARID2 KO lines. Densitometric measurements corresponding to the experiment are shown in Fig EV2D.

G   Immunoblot after a full-length JARID2 isoform-1 was expressed from an exogenous vector in HaCaT cells showed an increase (2- to 3-fold) in ~80 kDa level. Densitometric measurements are shown in Fig EV2E. An immunoblot with exogenous expression of 140kDa band in a control cell line is shown in Fig EV2F.

H   Immunoprecipitation using anti-JARID2 antibody and followed by mass spectrometry identification of the ~80 kDa band detected JARID2 peptides spanning only C-terminus of JARID2. The identified peptides are shown as red arrows on schematic of JARID2 sequence.

Source data are available online for this figure.

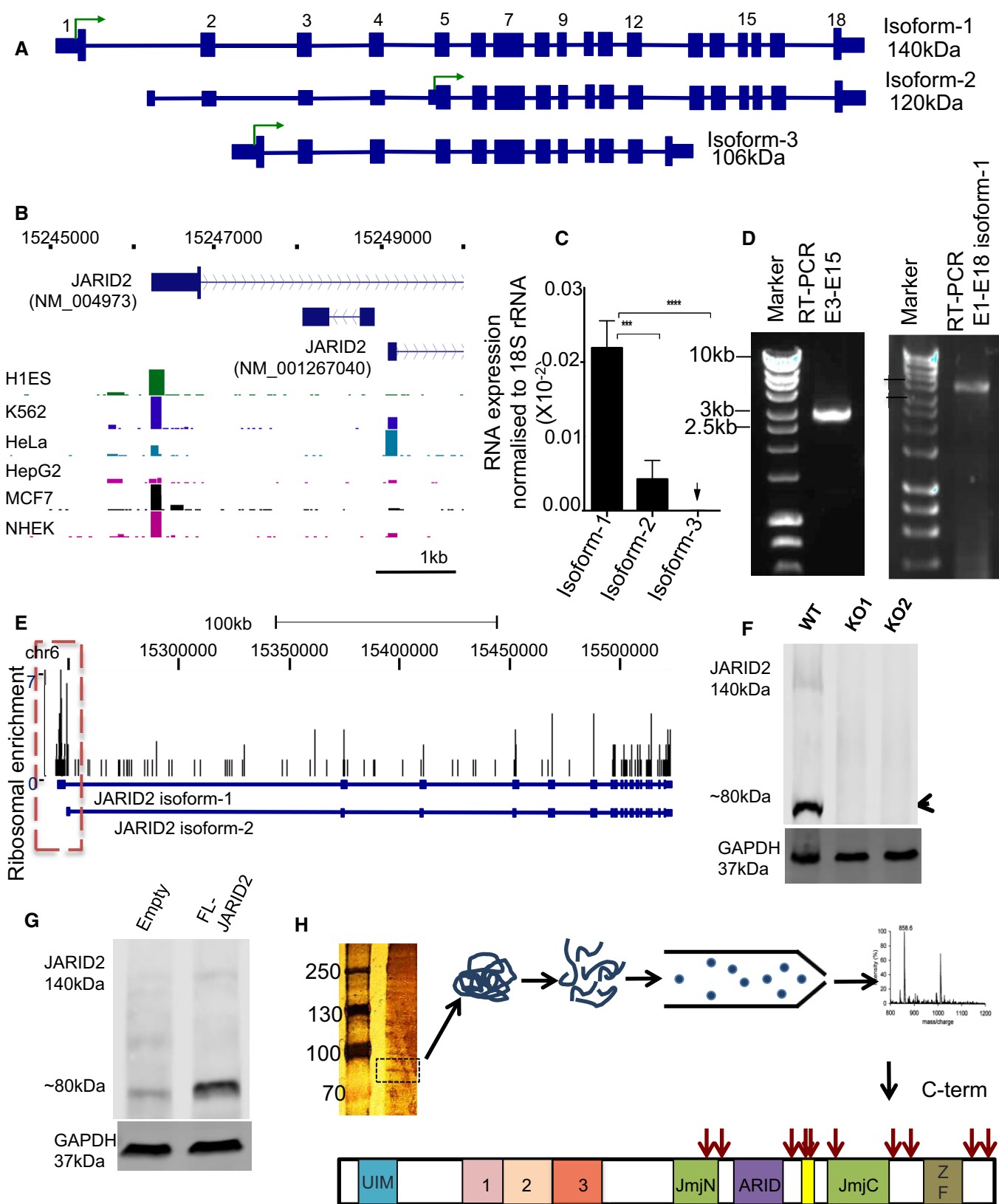

**Figure 2.**

spectrometry data where the N-terminal-most peptide is detected at amino acid position 589 of JARID2 sequence (Fig EV3).

### ΔN-JARID2 is required for cell differentiation

Previous studies have shown that PRC2 and JARID2 play an important role in epidermal development and differentiation (Ezhkova *et al*, 2009, 2011; Mejetta *et al*, 2011; Wurm *et al*, 2015). Therefore, we explored the role of ΔN-JARID2 in epidermal differentiation. First, we examined expression of ΔN-JARID2 in keratinocytes (HaCaTs), which represent a good model for human epidermal differentiation (Wilson, 2014). We differentiated HaCaTs by growing them to confluency in low calcium medium for 4 weeks before switching to high calcium medium for up to 6 days (Fig 3A). We confirmed differentiation by measuring levels of epidermal differentiation markers such as involucrin (IVL) and trans-glutaminase1 (TGase1) on days 0, 1, 3 and 6 after switching to high calcium medium (Fig 3B). Interestingly, as differentiation progressed, ΔN-JARID2 levels also increased up to 2-fold (Fig 3C). Therefore, we speculated that ΔN-JARID2 might be important for HaCaT differentiation.

To test this hypothesis, we compared differentiation in JARID2-null HaCaTs and wild-type HaCaTs by comparing RNA levels of differentiation markers keratin-1, keratin-10 and involucrin on day 0 and day 3 of differentiation (Fig 3D and E). We predicted that removal of JARID2 should result in de-repression of differentiation genes. However, in JARID2-null cells, we observed significant down-regulation of involucrin and keratin-1 genes, which are earlier reported to be polycomb targets in Keratinocytes (Sen *et al*, 2008) (Fig 3D and E). Given that ΔN-JARID2 is the main form of JARID2 in keratinocytes and its level increases during differentiation, we speculated that the impaired differentiation seen in JARID2-null cells is due to removal of the ΔN-JARID2 form rather than the full-length form of JARID2. In such a case, exogenous expression of ΔN-JARID2 should be sufficient to rescue down-regulation of differentiation markers. We therefore transfected JARID2-null cells with plasmids expressing the C-terminal ~80 kDa fragment (Appendix Fig S1). Since we are yet to identify exact cleavage site of JARID2, this C-terminal fragment should mimic the ΔN-JARID2 form. Significantly, expression of the C-terminal ~80 kDa fragment was sufficient to rescue the expression of differentiation markers, indicating that the effect of JARID2 knockout on differentiation is most likely due to ΔN-JARID2 (Fig 3D and E). As wild-type cells express low levels of full-length JARID2, it could be argued that the impaired differentiation in JARID2 KO might be a combined effect of both full-length and ΔN-JARID2. To rule out this possibility, we also studied the effect of exogenously expressed full-length JARID2 (FL-JARID2). However, on day 3, expression of full-length JARID2 leads to suppression of differentiation markers (Fig 3E). This supports a role for ΔN-JARID2 in promoting differentiation, whereas full-length JARID2, like other polycomb proteins, functions to suppress differentiation.

### ΔN-JARID2 is required for genome-wide up-regulation of differentiation genes

To understand the effect of JARID2 knockout on genome-wide expression patterns in HaCaTs, we carried out RNA-sequencing

analysis. Using RNA-seq data, we compared changes in gene expression patterns in JARID2-null vs JARID2-expressing wild-type HaCaT cells. We identified 645 genes which changed expression by more than 2-fold ($P < 0.0001$) between wild-type and JARID2-null cells which included 269 up-regulated genes and 376 down-regulated genes.

To further understand the functions of JARID2-regulated genes, we investigated the biological pathways these genes are involved in using Database for Annotation, Visualization and Integrated Discovery (DAVID; Jiao *et al*, 2012). We observed that several genes involved in DNA replication and G1/S transition of cell cycle were up-regulated in JARID2 KO cells (Appendix Fig S2A). On the other hand, several developmental and epidermal differentiation genes were down-regulated as compared to wild-type cells (Appendix Fig S2B).

This effect was much more pronounced when gene expression was profiled after 3 days of differentiation in wild-type (WT) and KO cells. After 3 days of differentiation, in wild-type HaCaTs, epidermal differentiation genes were highly up-regulated and cell cycle genes were down-regulated, further validating our differentiation protocol (Appendix Fig S2C and D).

As compared to WT, in JARID2 KO cells epidermal differentiation genes, as well as genes involved in extracellular matrix organisation, were significantly down-regulated indicating significant impairment of differentiation in the absence of JARID2 (Fig 4A). In addition, cell cycle genes in the KO cells were expressed at higher levels indicating continued proliferation and inhibition of cell cycle exit (Fig 4B). In summary, gene expression changes (Fig 4A and B) upon JARID2 removal indicate that JARID2 is needed for up-regulation of differentiation genes in keratinocytes.

We therefore first checked if the down-regulated genes are indeed JARID2 targets. Analysis of genome-wide JARID2 Chromatin Immunoprecipitation (ChIP) data (Kaneko *et al*, 2014a) from human pluripotent cells showed enrichment for JARID2 binding in the down-regulated genes compared to up-regulated genes (Fig 4C). We also examined if these genes are targeted by polycomb proteins (Margueron & Reinberg, 2011). Interestingly, down-regulated genes have GC-rich promoters that are characteristic of polycomb target genes (Appendix Fig S3A; Mendenhall *et al*, 2010). In addition, a multi-gene plot of EZH2 enrichment at these genes showed that they are bound by EZH2 in human ES cells as well as in keratinocytes (Fig 4D) indicating that these genes are PRC2 targets. The gene expression changes we observed in JARID2 KO can be due to altered targeting of PRC2. In such a scenario, we expect up-regulated genes to be the PRC2 targets. However, as compared to down-regulated genes, up-regulated genes showed lower enrichment of JARID2 and PRC2 (Fig 4C and D). Making it less likely that the down-regulation of differentiation genes we observed in JARID2 KO is due to mere rearrangement or altered targeting of PRC2. As JARID2 is required for PRC2 recruitment (Landeira & Fisher, 2011; Holoch & Margueron, 2017; Chen *et al*, 2018) and ΔN-JARID2 lacks PRC2 interacting domain, we investigated the possibility that down-regulation of differentiation genes in keratinocytes lacking JARID2 might be a consequence of changes in PRC2 activity. Our co-immunoprecipitation experiment confirmed that PRC2 subunit, EZH2, interacts with full-length JARID2 but not with ΔN-JARID2 (Appendix Fig S3B and C). However, similar to previous publications, we did not observe any significant change in global H3K27me3 levels in

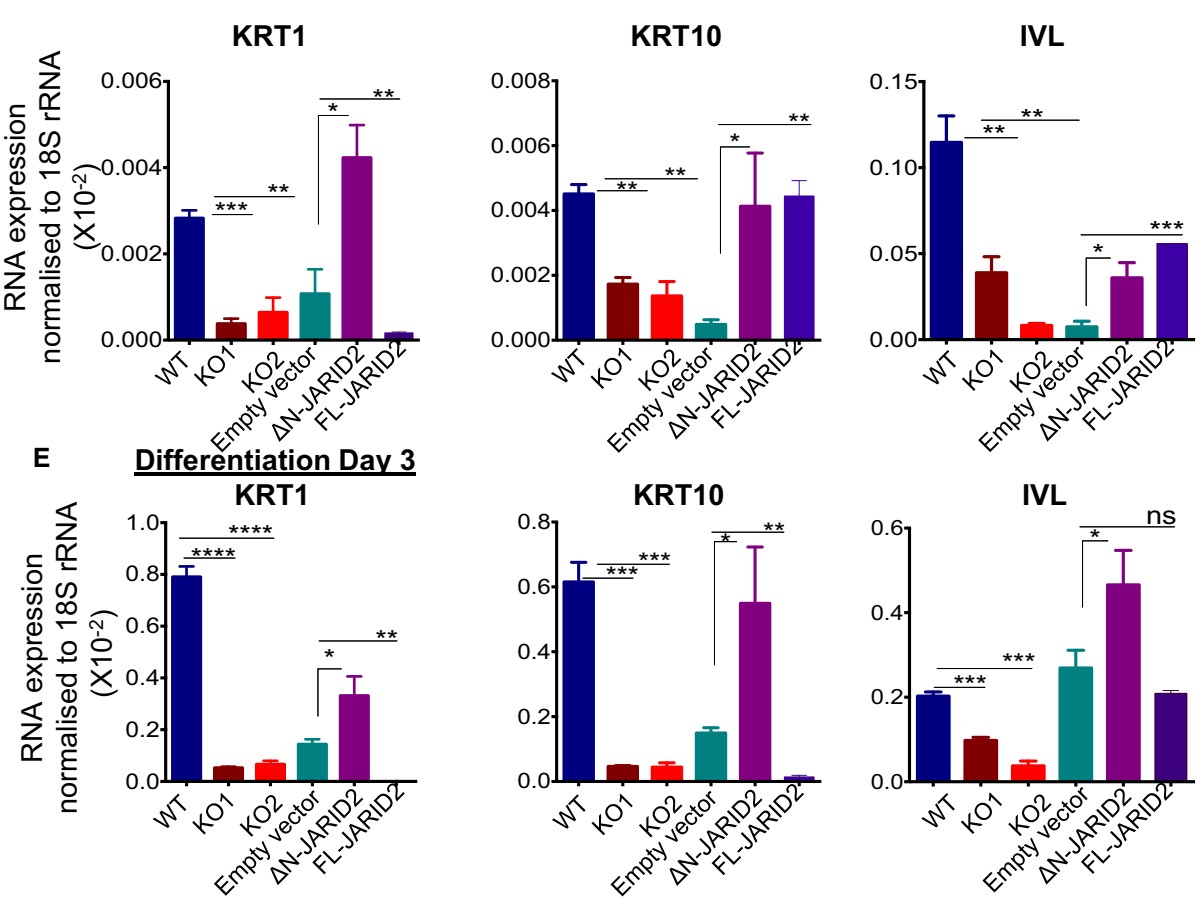

**Figure 3. Effect of JARID2 knockout on differentiation markers.**

A  The calcium-induced differentiation protocol used in this study. Up until day 0, cells were maintained in low calcium media and then induced to differentiate by growing them in high calcium medium for 6 days. Cells were harvested for immunoblot at day 0, day 1, day 3 and day 6 of differentiation.

B  Immunoblot showing increase in expression levels of differentiation markers involucrin (IVL) and Transglutaminase-1 (TGase-1) as differentiation progressed indicating that keratinocyte differentiation protocol was successful. Protein levels on day 0 (D0), day 1 (D1), day 3 (D3) and day 6 (D6) of differentiation are shown.

C  Immunoblot for JARID2 during D0, D1, D3 and D6 of differentiation (as in B) are shown.

D  Effect of JARID2 removal on levels of differentiation markers involucrin (IVL), keratin-1 (KRT1) and keratin-10 (KRT10) mRNAs as measured in qPCR experiment relative to 18S rRNA and the rescue using exogenous expression of ΔN-JARID2. The effect of exogenous expression of full-length JARID2 (FL-JARID2) in JARID2 knockout is also shown. Data from wild-type HaCaTs, two independent JARID2 knockout (KO1 and KO2) HaCaT lines and KO cells exogenously expressing empty vector control, ΔN-JARID2 and FL-JARID2 are shown. Data from three independent experiments ($n = 3$) are represented as mean ± SE, and multiple comparisons were performed using one-way ANOVA (****$P < 0.0001$, ***$P < 0.001$, **$P < 0.01$ and *$P < 0.05$).

E  Levels of involucrin (IVL), keratin-1 (KRT1) and keratin-10 (KRT10) mRNAs as above but measured on day 3 of differentiation.

Source data are available online for this figure.

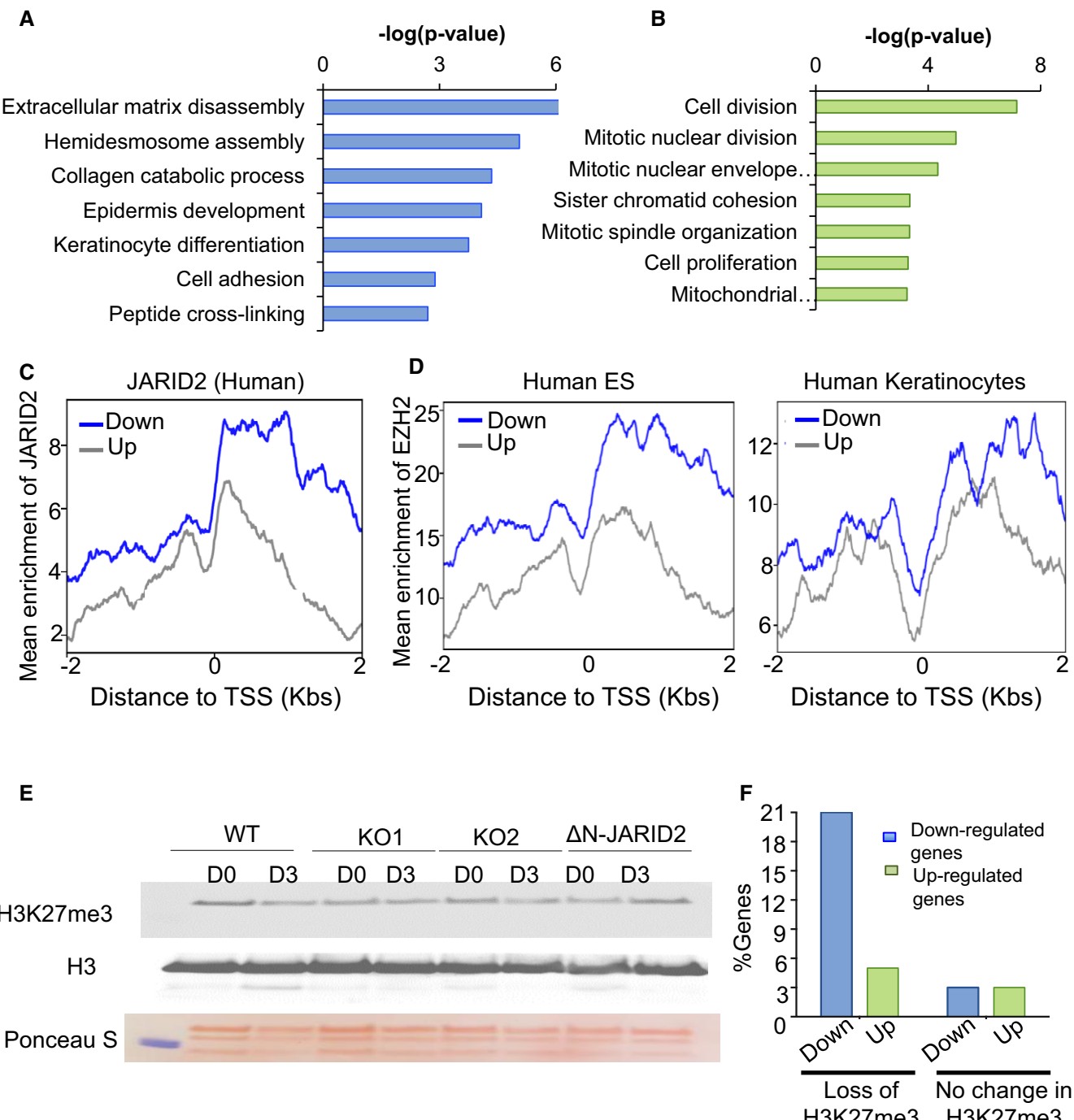

**Figure 4. Gene expression changes in JARID2-null cells.**

A, B   Functional enrichment of (A) 2-fold down-regulated and (B) 2-fold up-regulated genes in JARID2-null cells as compared to WT on day 3 of differentiation. Functional categories are plotted on vertical axes and −log(p) values are plotted on horizontal axes.

C      A metagene plot showing enrichment of full-length JARID2 in human-induced pluripotent cells at down-regulated genes (blue) as compared to up-regulated genes (grey) in JARID2 KO cells. The plot is centred on Transcription Start Sites (TSS) of genes and distance from TSS is indicated on the x-axis.

D      Metagene plots of average enrichment of EZH2 in ES cells and keratinocytes at down-regulated genes (blue) as compared to up-regulated genes (grey) in JARID2 KO cells. The plots are centred on Transcription Start Sites (TSS) of genes and distance from TSS is indicated on the x-axes.

E      Levels of H3K27me3 in WT, JARID2 KO lines and JARID2 KO lines expressing ΔN-JARID2. Histone H3 and Ponceau staining is used as a loading control. Densitometric measurements corresponding to this experiment are shown in Fig EV4A.

F      A bar-plot showing percentage of down- and up-regulated genes among genes which lose H3K27me3 vs which do not lose H3K27me3 modification during differentiation.

Source data are available online for this figure.

JARID2 KO or ΔN-JARID2 expressing cells (Figs 4E and EV4A). Although this method cannot rule out the possibility of altered targeting of PRC2, enrichment of PRC2 at down-regulated genes rather than up-regulated genes in keratinocytes (Fig 4D) implies an alternative scenario. In addition, we found that EZH2 levels in wild-type HaCaTs and JARID2 KOs are very similar (Fig EV4B), ruling out the possibility that the changes in gene expression pattern in JARID2 KO cells are simply result of changes in EZH2 expression. An alternative explanation is that ΔN-JARID2 is required for activation of differentiation genes. Our data (Fig 3D and E) support this hypothesis. We also observed that those epidermal genes that lose the H3K27me3 mark upon differentiation (Sen *et al*, 2008) are highly enriched in down-regulated genes. 4.2% of down-regulated genes lose H3K27me3 upon day 3 of differentiation as compared to only 1.6% of up-regulated genes and 1.4% of all genes (Fig 4F). This suggests that JARID2 is mainly required for activation of genes that lose their H3K27me3 mark.

Since ΔN-JARID2 contains conserved domains that are typical of the jumonji family of demethylases, we explored a possibility that at these genes ΔN-JARID2 helps in removal of the H3K27me3 mark. In such a scenario, inhibition of EZH2 enzymatic activity should have the opposite effect to that of JARID2 knockout and we should see up-regulation of differentiation genes. We therefore treated wild-type cells expressing JARID2 with EZH2 inhibitor UNC1999. Following 24-h treatment with UNC1999, H3K27me3 was reduced to very low level (Fig EV4C). However, this did not result in activation of epidermal differentiation genes (Fig EV4D). This is in agreement with a previous publication, which showed that, in keratinocytes, removing EZH2-mediated repression is not enough and additional factors are required for activation of polycomb targeted epidermal genes (Wurm *et al*, 2015). Another possibility is that ΔN-JARID2 is required for activation of transcription by aiding recruitment of certain transcription factors (Wurm *et al*, 2015) or RNA pol II (Landeira *et al*, 2010) or both. Indeed, an analysis of promoter sequences of down-regulated genes showed significant enrichment of motif of AP-1 transcription factor FOSL (Fig EV4E), which is needed for activation of epidermal differentiation genes (Wurm *et al*, 2015). Moreover, we did not observe enrichment of AP-1 motif in up-regulated genes indicating that down-regulated genes are targets of AP-1. This suggested that ΔN-JARID2 might aid AP-1-mediated up-regulation of differentiation genes. Interestingly, previous publication shows that interplay between EZH2 and AP-1 transcription factors is needed for keratinocyte differentiation (Wurm *et al*, 2015). EZH2-mediated methylation of AP-1 factor FOSL-2 inhibits its activity, which is relieved upon differentiation. However, we could not detect a direct interaction of either form of JARID2 with AP-1 (Appendix Fig S4A and B) suggesting that ΔN-JARID2 might not directly interact but might indirectly aid in activity of AP-1 transcription factors by relieving their EZH2-mediated repression.

## Discussion

It is well established that the N-terminal domain of JARID2 is needed for its interaction with EZH2 and its recruitment to chromatin (Son *et al*, 2013; Kaneko *et al*, 2014a; Sanulli *et al*, 2015; Cooper *et al*, 2016; Chen *et al*, 2018). By contrast, the C-terminal of

JARID2 has not been studied in detail. One of the reasons for this is that it has been shown to be unnecessary for EZH2 enzymatic activity or binding (Son *et al*, 2013; Kaneko *et al*, 2014a; Cooper *et al*, 2016). This is despite the fact that C-terminal contains several conserved domains typical of the jumonji family, as well as the zinc finger and ARID domains that are involved in DNA binding (Li *et al*, 2010).

In this study, we discovered that JARID2 exists as an ~80 kDa low molecular weight form that consists of the C-terminal domain. This low molecular weight form (which we termed ΔN-JARID2) is a cleavage product of full-length JARID2 that lacks PRC2-interacting domains and is the predominant form in many lineage-committed human cells including keratinocytes. This is a significant finding as the presence of another form of JARID2 can provide answers to multiple unsolved questions related to JARID2's association with polycomb function.

In keratinocytes, we could mainly detect C-terminal portion (or ΔN-JARID2) of JARID2, but its N-cleaved portion was missing. This suggests that N-terminal portion is not very stable in these cells. This can explain the discrepancy in JARID2 mRNA level and ΔN-JARID2 protein levels in lineage-committed cells. In these cells, JARID2 mRNA is expressed at detectable but albeit low levels, while ΔN-JARID2 protein levels are much higher. It can be hypothesised that the cleavage of unstable N-terminal portion from full-length JARID2 can make its C-terminus more stable leading to higher levels of ΔN-JARID2 than expected from JARID2 mRNA levels.

We find that during keratinocyte differentiation ΔN-JARID2 levels increase. This can explain the reported decrease in JARID2 full-length levels during differentiation (Zhang *et al*, 2011; Sanulli *et al*, 2015), which is likely due to increased cleavage of isoform-1 in addition to a decrease in RNA levels (Shen *et al*, 2009). We show that in human keratinocytes, JARID2 removal leads to suppression of PRC2 targets such as differentiation genes. Although consistent with previous observations (Shen *et al*, 2009; Landeira *et al*, 2010), this is puzzling, as it is expected that JARID2 knockout should interfere with PRC2 targeting and hence lead to de-repression of its target genes as observed in case of removal of other PRC2 components (Azuara *et al*, 2006; Boyer *et al*, 2006; Pasini *et al*, 2007; Shen *et al*, 2008). We show that decreased expression of epidermal differentiation genes seen in JARID2-null cells can be reversed by expression of the C-terminal fragment similar to ΔN-JARID2. This raises the exciting possibility that ΔN-JARID2 is needed for activation of differentiation genes. Enrichment of AP1 transcription binding motifs at JARID2 regulated genes suggests that ΔN-JARID2, directly or indirectly, might aid in recruiting transcriptional activators such as AP-1 at these genes. This is consistent with the previous observation, which shows that JARID2 is required for RNA pol II recruitment during differentiation (Landeira *et al*, 2010).

This is important as identification of ΔN-JARID2 and its role in activation of differentiation genes implies that JARID2 might function in two ways. In its full-length form it acts as a transcriptional co-repressor that functions through its interaction with PRC2 whereas in its cleaved form it acts as an activator for PRC2 target genes. We speculate that this might be a general regulatory mechanism of PRC2 components, as distinct molecular weight forms have been also reported in the case of AEBP2 (Kim *et al*, 2015), another subunit of JARID2 containing PRC2 complex. The presence of

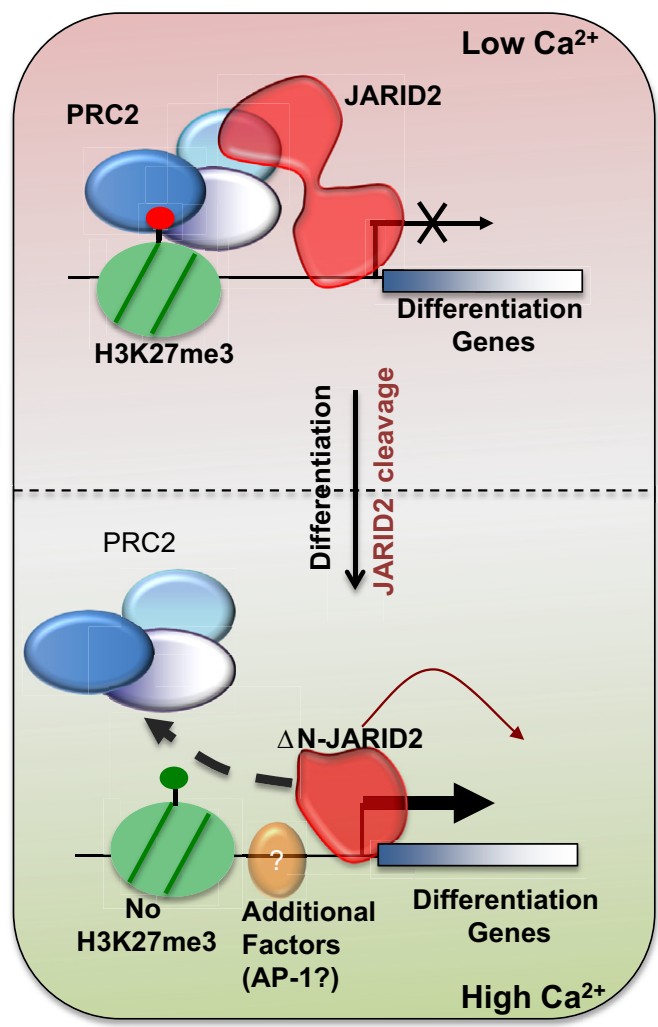

**Figure 5.  A model predicting the role of ΔN-JARID2 in differentiation.**
Jarid2 is present as a full-length protein in multi-potent and undifferentiated cells. Full-length JARID2 with the help of N-terminal PRC2 interacting domain can recruit PRC2 and suppress differentiation genes. Upon differentiation N-terminal of JARID2 is cleaved (ΔN-JARID2) which leads to reduction in PRC2 recruitment and possibly H3K27me3 levels at differentiation genes. However, PRC2 removal is not sufficient for activation of differentiation genes and, in addition, ΔN-JARID2 is required for their up-regulation likely through, directly or indirectly, facilitating transcription factors such as AP-1.

another form of JARID2 can also provide additional insight into how polycomb target genes are activated during differentiation.

It is known that PRC2 preferentially binds to GC-rich regions. However, it is not clear why PRC2 does not target CpG-rich promoters at active genes. Given that the C-terminal of JARID2 also has a preference for GC-rich sites (Mendenhall *et al*, 2010), it is possible that ΔN-JARID2 form might bind to CpG-islands at active genes and protect them from PRC2-mediated repression (Jermann *et al*, 2014; Riising *et al*, 2014). Recently, it was shown that EZH2 binds to RNA and nucleosomes in a mutually exclusive manner (Beltran *et al*, 2016). However, it is not clear what triggers EZH2's release from chromatin. It can be imagined that cleavage of JARID2 might lead to release of EZH2 from chromatin. But how JARID2 is cleaved? Is not entirely clear and needs to be studied further. However, strong

enrichment of AP-1 motif in the promoters of down-regulated genes in JARID2-null cells suggests that directly or indirectly ΔN-JARID2 plays a role in AP-1-mediated gene regulation. Therefore, we propose a model (Fig 5) where, upon differentiation, JARID2 is cleaved leading to removal of PRC2 and probably H3K27me3 at subset of genes. The resulting ΔN-JARID2 form, directly or indirectly, aids in recruitment of transcriptional activators such as AP-1 resulting in up-regulation of differentiation genes. However, the relation between AP-1 transcription factors and ΔN-JARID2 will need further investigation.

## Materials and Methods

### Cell culture

A spontaneously immortalised human keratinocytes cell line HaCaT and human embryonic kidney cells HEK-293T were grown in Dulbecco's modified Eagle's medium (DMEM, Gibco) supplement with 5% foetal bovine serum (FBS) and 1% penicillin-streptomycin (10,000 U/ml) in 5% humidified $CO_2$ incubator at 37°C. To maintain HaCaT cells in undifferentiated phenotype as previously described (Wilson, 2014), HaCaT cells were cultured in DMEM with 10% $Ca^{2+}$-chelated FBS, 200 mM L-Glutamine and 0.03 mM calcium at 75% confluence. For keratinocyte differentiation, HaCaT cells were grown at full confluency in high calcium (2.8 mM) growth medium for the indicated time points.

### Transfections

JARID2 siRNAs targeting exon 3 and exon 15 were obtained from Sigma. $2 \times 10^5$ HaCaT cells were transiently transfected with 80 pmol siRNAs using Lipofectamine RNAiMAX (Invitrogen) for 48 h. Non-silencing control (NSC) was used as control. HEK-293T were transiently transfected as 3:1 ratio of reagent to DNA using X-tremeGENE9 DNA transfection reagent (Roche). Full-length JARID2 was cloned in pEF6 vector with N-terminal flag-tag, and shorter fragments of JARID2 (Appendix Fig S1A) were cloned in pcDNA3.1 vector. Stable HaCaT cells were generated by transfecting cells with 5 μg JARID2 expression constructs using Amxa Nucleofector kit V (Lonza) for 24 h. After 24 h, medium was refreshed. Stable cells were selected by growing them in DMEM medium containing 10 μg/ml of Blasticidin or 500 μg/ml Geneticin antibiotics. The expression of the construct was tested by immunoblotting.

### CRISPR/Cas9-mediated genome editing

sgRNA targeting JARID2 was designed using Wellcome Trust Sanger Institute Genome Editing database (WGE) (Hodgkins *et al*, 2015). sgRNA was selected with minimum off-target effects and close to translational start site (ATG) of isoform-1. HaCaT cells were electroporated with 5 μg pX459 vector (Addgene) containing sgRNA targeting JARID2 using Amaxa Cell Line Nucleofector kit V (Lonza) for 24 h. Transfected cells were selected for 3 days in a medium containing 0.5 μg/ml puromycin. After that cells were then maintained in DMEM for 5 days. Selected cells were serially diluted to single cells and were let to grow till colonies were grown. Homozygous mutations were confirmed by amplifying targeted loci using

RT–PCR. RT–PCR products were cloned into pJET1.2 blunt vector (ThermoFisher Scientific), and at least 10 bacterial colonies were picked up for genotyping.

## Immunoblotting

Cells were lysed with 20 mM Tris pH 7.5, 150 mM NaCl, 0.5% Deoxycholic acid, 10mM EDTA and 0.5% Triton X-100 containing protease inhibitor cocktail (complete ULTRA tablets, Roche). Histones were extracted from cells by acid extraction method (Halsall *et al*, 2015). According to standard procedure, lysates were treated with loading buffer, separated by 12–10% SDS–PAGE, transferred onto nitrocellulose membrane (Bio-Rad) using Trans-Blot Turbo Transfer System (Bio-Rad) and immunoblotted with following primary antibodies: JARID2 (1:1,000, Cell Signaling Technology, USA; 1:1,000, GTX129020, GeneTex), EZH2 (1:1,000, Cell Signaling Technology, USA), involucrin (1:1,000, Sigma), transglutaminase-1 TGase-1 (1:1,000, Santa Cruz Biotechnology, INC), c-Jun (1:1,000, Cell Signaling Technology, USA) and H3K27me3 (1:1,000, 07-449 Millipore). GAPDH (1:1,000, ThermoFisher Scientific) and C-terminus of histone H3 (1:5,000, Abcam) are used as loading controls. Immunoblots were visualised using fluorescence detection and scanned using odyssey infrared detection system (LI-COR Biosciences). Densitometry analysis was done using Image Studio Lite (LI-COR Biosciences).

## Immunofluorescence

Cells were cultured on cover slips and were fixed in PBS containing 4% (w/v) paraformaldehyde for 10 min at RT. The fixed cells were permeabilised with 0.2% Triton X-100 in PBS for 3 min and then blocked with 5% BSA for 1 h. Cells were incubated with 6X His antibody overnight (1:500, Thermofisher) at 4°C, washed three times using PBS and incubated with Fluorescein (FITC)-AffiniPure donkey anti-Mouse IgG secondary antibody (Jackson Immuno-Research Laboratories Inc) for 1 h. Coverslips were washed thoroughly, mounted using Vectorshield with DAPI and analysed using a Nikon A1R confocal microscope.

## RNA-sequencing and metagene analyses

RNA-sequencing was carried out on RNA extracted from wild-type HaCaT cells and JARID2-null HaCaT cells at day 0 and day 3 of differentiation. The sequencing was carried out on three biological triplicates. RNA-sequencing libraries were prepared using TrueSeq method. All the libraries were paired-end sequenced on an Illumina HiSeq 2500 machine (University of Birmingham). Sequences were quality filtered and trimmed using cutadapt. The reads were mapped using Tophat package (Trapnell *et al*, 2009) against human genome (hg19). Differential analysis was done using cuffdiff programme. Differential genes were identified using false discovery cut-off of $1 \times 10^{-5}$ and used for further analysis. Analysis of functional annotation was carried out using DAVID (Jiao *et al*, 2012). Metagene plots and heatmap were generated using DeepTools package (Ramirez *et al*, 2014). *De novo* motif finding was carried out using Homer software (Heinz *et al*, 2010). Analysis of H3K27me3-positive genes in HaCaTs was carried out using previously published data (Sen *et al*, 2008).

## Co-Immunoprecipitation

HEK-293T cells were transfected with Empty vector (Control), Flag-tagged full-length JARID2 and ΔN-JARID2 vectors. After 72 h of transfection, protein was extracted from all sample. For each IP, protein G-coated magnetic Dynabeads® were suspended and incubated with desired antibody (1–10 μg). After 10-min incubation with antibody, beads were washed and Dynabeads®-Antibody complex was incubated with protein samples. After washing the beads, proteins were eluted in elution buffer and SDS sample buffer and loaded on standard SDS–PAGE gel along with 5% whole-cell extract. The presence of co-immunoprecipitated proteins was verified using immunoblotting with respective antibodies.

## Mass Spectrophotometric protein Identification

The immunoprecipitation of JARID2 was carried out using monoclonal anti-JARID2 antibody (Cell Signalling Technology, USA) as mentioned in co-immunoprecipitation protocol. The eluted protein sample was separated using an SDS–PAGE and silver stained. 80 kDa band was cut and peptides were identified using the Q-Exactive HF mass spectrophotometer.

## Statistical analysis

Result analysis was performed using GraphPad Prism version 6 software. Data were represented as mean ± SE of three independent experiments. Student's *t*-test was used to compare two groups. Multiple comparisons were done using one-way ANOVA. A *P*-value of < 0.05 was considered significant.

# Data availability

The raw and processed RNA-seq data are deposited in GEO database (accession no. GSE102116; https://www.ncbi.nlm.nih.gov/geo/query/acc.cgi?acc=GSE102116). EZH2 metagene plots and H3K27me3 heatmap were generated using ES and Keratinocyte data deposited in ENCODE database (GEO accession no.: GSE29611; https://www.ncbi.nlm.nih.gov/geo/query/acc.cgi?acc=GSE29611). JARID2 metagene plots were created using previously published dataset (GEO accession no: GSM1180131; https://www.ncbi.nlm.nih.gov/geo/query/acc.cgi?acc=GSM1180131).

**Expanded View** for this article is available online.

## Acknowledgements

We thank Dr C. Murphy, Prof. C. Bunce, Dr M. Tomlinson and Dr R. Jenner for useful discussions. We are also grateful to Dr D. Cunningham for help with mass spectrometry data analysis. AK is funded by SSfH fellowship from University of Birmingham. DA is supported by IDB funding. Part of this work was funded by Wellcome Trust ISSF grant.

## Author contributions

A.K. conceived the study and wrote the manuscript. D.A.-R., R.J., S.W., M.P., S.R., J.H. and A.K. carried out the experiments. D.A.-R. and R.J. helped in making figures and writing Methods section. N.A.H. contributed to design of experiments and writing of the manuscript.

## Conflict of interest

The authors declare that they have no conflict of interest.

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
