## [Review Process File · The EMBO Journal]

A Novel Form of JARID2 is Required for Differentiation in Lineage-Committed Cells

Diaa Al-Raawi, Rhian Jones, Susanne Wijesinge, John Halsall, Marija Petric, Sally Roberts, Neil A. Hotchin, Aditi Kanhere.

Review timeline:

Submission date:	19 th October 2017
Editorial Decision:	17 th November 2017
Revision received:	30 th June 2018
Editorial Decision:	17 th July 2018
Revision received:	15 th October 2018
Accepted:	25 th October 2018

Editor: Anne Nielsen

Transaction Report:

1st Editorial Decision

17th November 2017

Thank you for submitting your manuscript for consideration by the EMBO Journal. It has now been seen by three referees whose comments are shown below. As you will see from the reports, all three referees express interest in the findings reported in your manuscript but also raise a number of important concerns that you will have to address before they can support publication here.

Should you be able to address these criticisms in full - and provided that the original conclusions still hold true - we would be happy to consider a revised manuscript.

REFeree REPORTS

Referee #1:

In this manuscript entitled "A novel form of JARID2 is required for differentiation in Lineage-Committed Cells", the authors provide evidence of the existence of a lower molecular weight form of the JARID2 protein in lineage committed cells. This shorter, lineage committed form of JARID2 (Δ N-JARID2) excludes the N-terminal PRC2 binding domain. The existence of a lineage specific form of JARID2 itself is exciting and raises many interesting questions about its biology, such as, at what stage in development does it arise and what is its function. The authors address this showing a role for Δ N-JARID2 during keratinocyte differentiation using the HaCat transformed aneuploid immortal cell line. To do this, they perform knock out and rescue experiments as well as RNAseq and ChIP-seq analyses to provide evidence that this shorter form of JARID2 is required for differentiation.

Major points:

1. To strengthen the experiment in Figure 1, they should test at least two different JARID2 antibodies with a knock down experiment in additional cell lines.

2. To strengthen the overall paper, I recommend that they pool Figures 2 and 3, since much of the information in Figure 2 is negative data and could easily be moved to the supplementary information.
3. Figure 3C supports the possibility that exogenous expression of full length JARID2 increases the expression of Δ N-JARID2. However, in the cell line used, full length JARID2 is not expressed. Therefore, as a control, the authors should repeat this experiment in a cell line that expresses full length JARID2, e.g. ES and embryonic carcinoma cells.
4. The data mapping the cleavage site in Figure 3D-E is still preliminary. To determine the exact site of cleavage/truncation, the authors could use their C-terminal antibody and perform an IP-mass spec for both forms of JARID2. This would allow them to map the peptides to each other to determine the exact cleavage site.
5. Figure 4 has many interesting aspects, namely the increase in expression of Δ N-JARID2 after differentiation and the ability to rescue the KO phenotype with exogenous expression of Δ N-JARID2. However, the authors should perform ChIP-qPCRs for long and short JARID2 at these genes in ES cells and in HaCat cells. This would allow them to determine if there is a shift from one form to the other during differentiation.
6. To support their model that Δ N-JARID2 replaces full length JARID2 and displaces PRC2 from chromatin during differentiation, the authors should ChIP PRC2 core complex components (EZH2, SUZ12 and EED), as well as JARID2 (short and long), on differentiation genes before and after induction of differentiation.

Minor points:

1. In Figure 1C, the authors nicely show a knock down of the 80kDa band. Here the authors should also show the 140kDa JARID2 region of the gel to control for full-length JARID2.
2. In the figure legends of Figures 2D and 2E, the authors should clarify if the PCR is an RT-PCR. They state this in the text but it is not clear in the figure or accompanying legend.
3. They should more carefully work out the mRNA transcript(s) from the JARID2 gene locus by either/both performing Northern blots or mapping each exon specifically using multiple exon specific primers in RT-PCRs.
4. The authors should indicate within the figures panels the cell type that is represented.
5. As a control for Figure 4D and E, the authors should exogenously express full length JARID2 and evaluate if it has the same effect (or not).
6. To achieve a semi-quantitative account of the western analysis in Figure 5E, the authors should blot a dilution series of their lysates

Referee #2:

This is a well presented and interesting study that reports a novel truncated C-terminal variant of JARID2, derived from previously unrecognised protein cleavage event. Interestingly the report highlights that unlike the parent full-length JARID2 peptide, the cleaved C-terminal JARID2 peptide is present in lineage committed human cell lines. Furthermore, they demonstrate that expression of a peptide approximate to Δ N-JARID2 in JARID2 null keratinocytes rescues expression of epidermal differentiation genes, and propose a switch to Δ N-JARID2 as a mechanism for gene regulation during differentiation. The findings are novel and potentially highlight an important and previously unrecognised role for JARID2 in lineage committed cells, and could be interesting to a wider readership.

The main concern is how robustly it has been demonstrated that the 80kDa band corresponds to a truncated JARID2 fragment, as this finding is confirmed by indirect methods. We note that the data sheet for the JARID2 antibody (Cell Signalling Technology USA) used in this study demonstrates a band at 80kDa in addition to the predicted 140kDa band, however we have not seen this band on data sheets for other C-terminal specific JARID2 antibodies. As this is a potentially highly significant result it is important to validate this finding directly, with a mass spectrometry based approach (ideally), and alternative C-terminal specific JARID2 antibodies (minimally), e.g. Abcam ab184152, raised against recombinant human JARID2 a.a.s 662-891.

Also the authors propose that the Δ N-JARID2 peptide is derived from the JARID2 isoform 1 mRNA. Do the authors have any knowledge of the factors that up-regulate expression of the full-

length form in differentiating cells, given the lack or very low-level detection of the parent 140kDa form of JARID2 in such cells? Can they show the presence of sufficient levels of mRNA encoding the full-length form in keratinocytes?

Major Issues

1. The Δ N-JARID2 peptide is identified by a single antibody, then confirmed by loss of immunoblotting following either siRNA targeting of the Jarid2 mRNA, or CRISPR-Cas9 mediated targeting of the JARID2 mRNA isoform 1 translational start site - i.e. indirect methods; how robust is this? Has this fragment been probed with alternate C-terminal JARID2 specific antibodies? Have the authors considered a Mass spec approach to confirm the identity of this band as a truncated JARID2 variant?
2. The authors elegantly demonstrate that there is no alternate mRNA transcript, splice variant or IRES, and conclude Δ N-JARID2 is therefore generated from the isoform 1 transcript, and subsequently cleaved from the full length protein. However it has previously been reported that JARID2 is not expressed in lineage committed cells, and expression atlases show generally low JARID2 transcript levels in differentiated tissues. How do they explain the apparent dichotomy of reported low Jarid2 mRNA expression, but high Δ N-JARID2 protein signal? Have they confirmed isoform 1 JARID2 mRNA expression in all the cell lines with qPCR (I can only see HaCaT cells, and it's not clear to me what the expression is relative to)? How do JARID2 expression levels in ES cells and the differentiating cell lines mentioned in the study compare?
3. There remains controversy as to the role of JARID2 in recruitment of PRC2 to chromatin targets. Specifically, Histone 3 lysine 27 trimethylation levels are varied across studies in JARID null ES cells. Furthermore there is recent evidence of a role for JARID2 in some differentiated cell types (e.g. 'Transcriptional Mechanisms Underlying Lymphocyte Tolerance': Macion et al, Cell 190(6), 719-731 (2002), 'Jarid2 is Induced by TCR Signalling and Controls iNKT Cell Maturation'; Pereira et al., Nat Commun. 5, 4540 (2014)). We feel that the authors could do more to broaden the introduction and highlight this controversy.
4. The authors speculate that Δ N-JARID2 'aids in recruitment of transcriptional activators such as AP-1 resulting in up regulation of differentiation genes', however this is based solely on the enrichment of the AP-1 binding motifs in the promoters of down-regulated genes in JARID2-null cells. We feel that the authors should provide experimental evidence of an interaction between Δ N-JARID2 and AP-1, or this should be referenced. Have the authors considered co-immunoprecipitation studies to demonstrate an interaction between JARID2 and AP1?

Minor Issues

1. The authors demonstrate that a peptide corresponding to amino acids 554-1246 is of equivalent size to the Δ N-JARID2 peptide, however they have not been able to confirm the exact nature of the peptide cleavage, do they have any thoughts on how to clarify this?
2. The authors subsequently use an approximate Δ N-JARID2 peptide to demonstrate that Δ N-JARID2 can rescue expression of differentiation markers in JARID2 null cells. It is important to recognise when interpreting these results that the reported Δ N-JARID2 has not been fully characterised, and the Δ N-JARID2 used for these experiments is an approximation based on peptide size determined by gel electrophoresis.
3. On page 13 - I think they mean lose, rather than loose?

Referee #3:

Summary

Al-Raawi et al. report that a C-terminal truncated form of JARID2, an accessory subunit of PRC2, is enriched in differentiated cells and activates gene expression via AP-1. The authors identify Δ N-JARID2 as an abundant 80kDa species in extracts from differentiated cell lines and confirm that this band corresponds to a low molecular weight form of JARID2 by RNAi. Analyses of CAGE data, isoform expression, and exon inclusion demonstrate that Δ N-JARID2 derives from cleavage of the full-length protein. Using an in vitro keratinocyte differentiation system, they show that Δ N-JARID2 has a role in differentiation, as overexpression of Δ N-JARID2 partially rescues marker expression in JARID2 KO cells. RNA-seq analyses showed that keratinocyte differentiation genes are down-regulated in JARID2 KO, and these genes are targets of JARID2 and EZH2 in mouse ES cells. Finally, the authors argue that JARID2 activates gene expression independently of its effects on PRC2 in keratinocyte differentiation, as no global change in H3K27me3 levels are observed upon

JARID2 KO and show that genes downregulated in JARID2 KOs are enriched for the AP-1 motif.

Critique

The regulation of PRC2 function by its various accessory subunit, including JARID2 is a topic of wide interest and the finding of a novel isoform of this protein with dedicated function in differentiated cells would be notable. However, the data as presented are insufficient to support the conclusions. Most importantly, I am not convinced that Δ N-JARID2 is actively produced in vivo and not a degradation product. Also the statement that the N-terminal domain of JARID2 is necessary for EZH2 interactions is in contradiction with the literature.

Major points

1) Existence of Δ N-JARID2 in vivo

1a: full length JARID2 is a large protein that is difficult to transfer on membrane by western blot due to its high pI (~9.5). An alternative interpretation of Fig. 1-3 is that the authors detect a degradation product that transfers more readily than the full length and is generated during or after lysis. Although the authors use suitable protease inhibitors this remains a possibility that should be formally excluded before proceeding to the functional dissection of this novel isoform.

1b: the authors claim that the Δ N-JARID2 form is enriched in "lineage-committed cells", yet their entire panel is made of transformed cancer cell lines (Fig. 1B). The relative abundance of Δ N- and FL-JARID2 should be measured and compared in pluripotent cells and primary differentiated cells.

1c: if Δ N-JARID2 is the result of an active proteolytic cleavage what is the fate of the N-terminal fragment? Is it degraded or does it remain intact (and presumably continues to interact with PRC2)? This could be easily addressed using the FLAG fusion mentioned in the text.

1d: the difference in the relative ratio of FL- and Δ N-JARID2 in Fig. 1B (lane 2) and Fig. 4C d0 (lane 1) is consistent with Δ N-JARID2 being a degradation product whose abundance might vary across extracts.

2) Δ N-JARID2 and PRC2

2a: The authors claim that a JARID2 fragment missing the first 500 aa does not interact with PRC2 but provide no data in support of this statement. They cite 4 studies (Refs. 16-19), of which only one supports their statement (Pasini et al.) Cooper et al. cite Pasini et al.; Son et al. show that the C-terminal fragment binds (albeit with reduced affinity) and Kaneko et al. cite Son et al. The authors omit to mention that in their ref. #14 PRC2 binding was mapped to the C terminus (Peng et al. Cell 2009, Fig. 2B). Given these contrasting reports, the lack of PRC2 binding of Δ N-JARID2 should be demonstrated in this manuscript.

2b: The authors infer that genes affected by loss of JARID2 in human keratinocytes are PRC2 targets based on distribution of PRC2 in mouse ESC cells. This should be confirmed in the relevant cell type or at least in the relevant species.

2c: If the authors wish to make the point that downregulated genes are disproportionately affected they should show in Fig. 5C and D the profile for the subset of upregulated genes not just the randomly selected control.

2c: In Pg. 13 the authors argue that lack of global changes in H3K27me3 in KOs demonstrate that PRC2 targeting and activity are not affected by the mutation. While global changes in activity would be detectable (although with low sensitivity) in a WB, altered targeting would not.

3) Transcriptional effects of Δ N-JARID2

3a) The authors suggest that all changes in gene expression in this model are due to loss of the truncated Δ N-JARID2 version but FL-JARID2 is also present in these cells and could contribute to the phenotype.

3b) The conclusion that Δ N-JARID2 is a transcriptional activator that acts via AP-1 is supported very indirectly by the data and would require much additional work to prove. I suggest that the authors tone down these conclusions or provide the additional data.

Minor points

- Fig. 1C: This blot should include the full-length JARID2 band to show effectiveness of the siRNA on the canonical isoform. What are the units for "relative expression"?

- Fig. 2B: To support this conclusion it should be shown that the KO mutants still express same levels of full length JARID2 mRNA.

- Pg. 9: The statement "in a blot with anti-Flag antibody, which will detect the N-terminal tag on

JARID2, we could not detect the LMW JARID2 form" should be supported by data.

1st Revision - authors' response

30th June 2018

Reviewer 1:

1. To strengthen the experiment in Figure 1, they should test at least two different JARID2 antibodies with a knock down experiment in additional cell lines.

As per reviewer's suggestion, we have now further analysed the 80kDa band using JARID2 siRNAs in additional cell lines (HEK293T, K562; revised Fig 1D, Fig EV1 A, B). In both the additional cell lines we could successfully knockdown the 80kDa band. These experiments also demonstrated the effectiveness of the two siRNAs against canonical JARID2 (140kDa), the knockdown of which can be clearly seen (revised Fig 1D, Fig EV1B).

In addition, we have also blotted the knockdown lysates using another C-terminal antibody from GeneTex (GTX129020), which targets a region of JARID2 spanning amino acids 662-891. This antibody also detects the 140kDa and 80kDa bands further supporting our finding that the 80kDa band corresponds to JARID2 (Fig EV1B).

2. To strengthen the overall paper, I recommend that they pool Figures 2 and 3, since much of the information in Figure 2 is negative data and could easily be moved to the supplementary information.

As per the reviewer's recommendation, we have combined figures 2 and 3 (Fig 2 in the revised version).

3. Figure 3C supports the possibility that exogenous expression of full length JARID2 increases the expression of Δ N-JARID2. However, in the cell line used, full length JARID2 is not expressed. Therefore, as a control, the authors should repeat this experiment in a cell line that expresses full length JARID2, e.g. ES and embryonic carcinoma cells.

We have carried out transfection experiments in another cell line, HEK293T, where transfection efficiency and plasmid expression are much higher when compared to HaCaTs. In these cells, we can see full-length JARID2 as well as high-level expression of Δ N-JARID2, indicating that our vector is expressing the right protein and can serve as a positive control (Fig EV2F).

4. The data mapping the cleavage site in Figure 3D-E is still preliminary. To determine the exact site of cleavage/truncation, the authors could use their C-terminal antibody and perform an IP-mass spec for both forms of JARID2. This would allow them to map the peptides to each other to determine the exact cleavage site.

We have now carried out an IP-mass spectrometry identification of Δ N-JARID2 and have mapped peptides for this form (Fig 2H, Fig EV3). All the peptides detected in the mass spectrometry identification (the majority with high confidence) spanned the C-terminus. The N-terminal most peptide we found in this analysis (starting at 589aa) is very close to our prediction of the cleavage site (554aa). Mass spectrometry identification depends on fragmentation efficiency of the protein, as well sampling of these peptides in the mass-spectrometer and, therefore, it is not possible to determine the exact cleavage site with this method.

5. Figure 4 has many interesting aspects, namely the increase in expression of Δ N-JARID2 after differentiation and the ability to rescue the KO phenotype with exogenous expression of Δ N-JARID2. However, the authors should perform ChIP-qPCRs for long and short JARID2 at these genes in ES cells and in HaCat cells. This would allow them to determine if there is a shift from one form to the other during differentiation.

We agree that it would be interesting to carry out these experiments. However, there are many technical challenges with this experiment. As our antibody recognizes C-terminus we would not be able to distinguish between the long and short form of JARID2. One way to solve this would be to carry out ChIPs using an N-terminal antibody and a C-terminal antibody. However, none of the available ChIP grade N-terminal antibodies are sufficiently specific (personal communication with Dr. Richard Jenner).

6. To support their model that Δ N-JARID2 replaces full length JARID2 and displaces PRC2 from chromatin during differentiation, the authors should ChIP PRC2 core complex components (EZH2, SUZ12 and EED), as well as JARID2 (short and long), on differentiation genes before and after induction of differentiation.

We agree that it would be informative to carry out ChIPs on EZH2, SUZ12 and EED, as well as JARID2 and we have attempted these experiments. Unfortunately, due to low levels of PRC2 proteins in HaCaTs (see figure B below) these experiments have not yielded usable data. Therefore, for success of these experiments, ChIP-procedure will need to be optimised in HaCaTs and will take much longer than what fits in the scope of this paper.

Figure 1: A) ChIP-qPCR at HOXB5, a polycomb target gene (positive control). The ChIP experiments were performed using IgG, SUZ12, EED and the JARID2 antibody recognizing C-term (CST) on samples from undifferentiated (WTd0) and differentiated (WTd3) HaCaTs as well as JARID2 knockout (KOd0). In majority ChIPs, we did not see any significant enrichment over IgG at HOXB5 gene which is a known target of polycomb.

B) A western blot showing levels of the polycomb protein, SUZ12, in whole cell extract (WCE) and ChIPed samples from Fig1A (above). Histone H3 was used as a loading control for WCE.

Minor points

In Figure 1C, the authors nicely show a knock down of the 80kDa band. Here the authors should also show the 140kDa JARID2 region of the gel to control for full-length JARID2.

As requested, we have now shown the 140kDa region of the western blot. As the reviewer pointed out in comment 3, in our cells we could detect only very low or undetectable levels of the 140kDa band. In addition, Fig 1C (Fig 1D in the revised MS) is a transient knockdown experiment and to achieve high transfection efficiency, these experiments were done on low number of cells explaining why we were unable to see the 140kDa band in these experiments. However, as mentioned in response to comment 1, we have repeated these experiments in additional cell-lines (revised Figure 1D, Figure EV1A and B) where the 140kDa band is visible, and is reduced following siRNA-mediated knockdown of JARID2.

In the figure legends of Figures 2D and 2E, the authors should clarify if the PCR is an RT-PCR. They state this in the text but it is not clear in the figure or accompanying legend.

We thank reviewer for pointing this out. We have revised this in the current version of the manuscript (Fig 2D and its legend).

They should more carefully work out the mRNA transcript(s) from the JARID2 gene locus by either/both performing Northern blots or mapping each exon specifically using multiple exon specific primers in RT-PCRs.

As mentioned in the text we can only amplify a single band when siRNA targeted exon 3 and exon 15 primers are used. We have now sequenced this band and confirmed presence of all the exons and gene structure. This data is provided in the Appendix.

The authors should indicate within the figures panels the cell type that is represented.

This has been included in revised version. In addition, we have also included additional data from ES cells and primary differentiated cells (revised Fig 1B).

As a control for Figure 4D and E, the authors should exogenously express full length JARID2 and evaluate if it has the same effect (or not).

As per the reviewer's suggestion, we have exogenously expressed full length JARID2 (FL-JARID2) and evaluated the effect of this on gene expression changes during differentiation. These data are included in Fig 3D and E. We do not see the same effect as when Δ N-JARID2 is expressed and, but interestingly, exogenous expression shows an effect similar to other polycomb proteins that keep differentiation genes repressed (discussed in text page 12, paragraph 2; page 13, paragraph 1).

6. To achieve a semi-quantitative account of the western analysis in Figure 5E, the authors should blot a dilution series of their lysates

All immunoblots were analysed using a Li-Cor Odyssey Infrared Imaging system. This has a much wider linear dynamic range than traditional film/chemiluminescence and we are confident the level of signal observed is within that linear range. To further address the reviewers concerns over quantitation, we have carried out densitometry analysis of 3 independent experiments. These data are shown in the supplementary information (Fig EV4A) and confirm there is no significant difference in global H3K27me3 in either JARID2 KO or DN-JARID2 expressing cells.

Reviewer 2**Major Issues**

1. The Δ N-JARID2 peptide is identified by a single antibody, then confirmed by loss of immunoblotting following either siRNA targeting of the Jarid2 mRNA, or CRISPR-Cas9 mediated targeting of the JARID2 mRNA isoform 1 translational start site - i.e. indirect methods; how robust is this? Has this fragment been probed with alternate C-terminal JARID2 specific antibodies? Have the authors considered a Mass spec approach to confirm the identity of this band as a truncated JARID2 variant?

We have now further analysed the C-terminal form of JARID2 using siRNA knockdown in two additional cell lines (HEK293T and K562; Fig 1D and Fig EV1A, B), an alternate C-terminal antibody (GeneTex GTX129020; Fig EV1B) and an IP-mass-spectrometry experiment (Fig 2H and Fig EV3 in the revised MS). All these experiments indicate that the 80kDa band is JARID2. In our mass-spectrometry experiment only C-terminal peptides are detected from the 80kDa band indicating it is a truncated JARID2 variant.

2. The authors elegantly demonstrate that there is no alternate mRNA transcript, splice variant or IRES, and conclude Δ N-JARID2 is therefore generated from the isoform 1 transcript, and subsequently cleaved from the full length protein. However it has previously been reported that JARID2 is not expressed in lineage committed cells, and expression atlases show generally low JARID2 transcript levels in differentiated tissues. How do they explain the apparent dichotomy of reported low Jarid2 mRNA expression, but high Δ N-JARID2 protein signal? Have they confirmed isoform 1 JARID2 mRNA expression in all the cell lines with qPCR (I can only see HaCaT cells, and it's not clear to me what the expression is relative to)? How do JARID2 expression levels in ES cells and the differentiating cell lines mentioned in the study compare?

This is a fundamental question. To address this, we have measured RNA levels in all human cell lines used in this study as well as in human ES cells (revised Fig 1C). HaCaTs and other lineage-committed cells show lower but detectable levels of mRNAs when compared to ES cells. The apparent dichotomy in the levels of mRNA and Δ N-JARID2 protein signal can potentially be explained by differences in stability of full-length JARID2 and Δ N-JARID2 (discussed in the main text: page 18, paragraph 1). Supporting this, our prediction analysis shows that the N-terminus of JARID2 is preferentially ubiquitinated. Therefore, its removal would make Δ N-JARID2 much more stable. However this is a subject of further study.

3. There remains controversy as to the role of JARID2 in recruitment of PRC2 to chromatin targets. Specifically, Histone 3 lysine 27 trimethylation levels are varied across studies in JARID null ES cells. Furthermore there is recent evidence of a role for JARID2 in some differentiated cell types (e.g. 'Transcriptional Mechanisms Underlying Lymphocyte Tolerance': Macion et al, Cell 190(6), 719-731 (2002), 'Jarid2 is Induced by TCR Signalling and Controls iNKT Cell Maturation'; Pereira et al., Nat Commun. 5, 4540 (2014)). We feel that the authors could do more to broaden the introduction and highlight this controversy.

Taking the reviewer's comments into account, we have revised the Introduction to include these points (Page 4; page 5 Paragraph 1).

4. The authors speculate that Δ N-JARID2 'aids in recruitment of transcriptional activators such as AP-1 resulting in up regulation of differentiation genes', however this is based solely on the enrichment of the AP-1 binding motifs in the promoters of down-regulated genes in JARID2-null cells. We feel that the authors should provide experimental evidence of an interaction between Δ N-JARID2 and AP-1, or this should be referenced. Have the authors considered co-immunoprecipitation studies to demonstrate an interaction between JARID2 and AP1?

Acting on the reviewer's suggestion, we have carried out co-immunoprecipitation experiments (Fig S4). In these experiments, we could not detect any interaction between JARID2 and the AP-1 protein c-Jun that we tested. However, a recent publication (Wurm et al. 2015 *Genes Dev.*), which is cited in our manuscript, showed that AP-1 plays an important role in terminal differentiation of Keratinocytes. They also showed that AP-1 complex binds to EZH2 target promoters. At these promoters, the AP-1 protein Fra-2 is methylated by EZH2 in undifferentiated cells and this renders the AP-1 complex inactive. However, upon triggering of differentiation, AP-1-EZH2 interaction is lost leading to loss of Fra2 methylation. This is followed by Fra2 phosphorylation, which leads to activation of differentiation genes. We have cited this study. Based on our co-immunoprecipitation experiments, we have fine-tuned our model to avoid any suggestion that Δ N-JARID2 recruits AP-1 (page 16, paragraph 2; page 17, paragraph 1; Page 18, paragraph 2; page 19, paragraph 1; page 20).

Minor Issues

1. The authors demonstrate that a peptide corresponding to amino acids 554-1246 is of equivalent size to the Δ N-JARID2 peptide, however they have not been able to confirm the exact nature of the peptide cleavage, do they have any thoughts on how to clarify this?

To help identify the JARID2 cleavage mechanism, we have identified a potential protease target site around residue 554. However, there are multiple enzymes that might act at this site and we feel that identifying the exact site and protease responsible for JARID2 cleavage is beyond the scope of this paper.

2. The authors subsequently use an approximate Δ N-JARID2 peptide to demonstrate that Δ N-JARID2 can rescue expression of differentiation markers in JARID2 null cells. It is important to recognise when interpreting these results that the reported Δ N-JARID2 has not been fully characterised, and the Δ N-JARID2 used for these experiments is an approximation based on peptide size determined by gel electrophoresis.

The text has been revised to make this point clear (Page 12, paragraph 2, lines 17-23).

3. On page 13 - I think they mean lose, rather than loose?

The text has been corrected to replace "loose" with "lose".

Reviewer 3

Major points

1) Existence of Δ N-JARID2 in vivo

1a: full length JARID2 is a large protein that is difficult to transfer on membrane by western blot due to its high pI (~9.5). An alternative interpretation of Fig. 1-3 is that the authors detect a degradation product that transfers more readily than the full length and is generated during or after lysis. Although the authors use suitable protease inhibitors this remains a possibility that should be formally excluded before proceeding to the functional dissection of this novel isoform.

To rule out the possibility of degradation we have carried out western blots using protein lysates extracted in presence of increasing amounts of protease inhibitor (Fig EV1C in revised MS) and we do not see any difference in the levels of full length or short form. In addition, in ES cells (Fig 1B) processed exactly the same way, we only see full-length JARID2, indicating that this is not a procedural artifact.

1b: the authors claim that the Δ N-JARID2 form is enriched in "lineage-committed cells", yet their entire panel is made of transformed cancel cell lines (Fig. 1B). The relative abundance of Δ N- and FL-JARID2 should be measured and compared in pluripotent cells and primary differentiated cells.

We appreciate this suggestion and we have now included human and mouse ES cells as well as primary differentiated cells in the panel (Fig 1B). We have also indicated the cell types in the figure.

1c: if Δ N-JARID2 is the result of an active proteolytic cleavage what is the fate of the N-terminal fragment? Is it degraded or does it remain intact (and presumably continues to interact with PRC2)? This could be easily addressed using the FLAG fusion mentioned in the text.

In HaCaTs we cannot detect the N-terminal flag-tagged fragment (Fig EV2G). According to our prediction of ubiquitination, the N-terminus of JARID2 should be much more unstable than C-terminus, which could explain why we do not see this fragment. We have included a short section in the discussion to reflect on this point (page 18, paragraph 1).

1d: the difference in the relative ratio of FL- and Δ N-JARID2 in Fig. 1B (lane 2) and Fig. 4C d0 (lane 1) is consistent with Δ N-JARID2 being a degradation product whose abundance might vary across extracts.

As mentioned above, we have carried out multiple tests to rule out the possibility of degradation. The cell conditions in Fig 1B (lane 2) are not the same as those in Fig 4C (lane 1; new Fig 3C). d0 cells (Fig 4C lane 1; new Fig 3C) are grown in low Ca^{2+} medium whereas the cells in Fig 1B are grown in standard DMEM media containing calcium. This has been now clarified in the text (page 11, paragraph 2; Fig 3A legend) and Fig 3A.

2) Δ N-JARID2 and PRC2

2a: The authors claim that a JARID2 fragment missing the first 500 aa does not interact with PRC2 but provide no data in support of this statement. They cite 4 studies (Refs. 16-19), of which only one supports their statement (Pasini et al.) Cooper et al. cite Pasini et al.; Son et al. show that the C-terminal fragment binds (albeit with reduced affinity) and Kaneko et al. cite Son et al. The authors omit to mention that in their ref. #14 PRC2 binding was mapped to the C terminus (Peng et al. Cell 2009, Fig. 2B). Given these contrasting reports, the lack of PRC2 binding of Δ N-JARID2 should be demonstrated in this manuscript.

As per reviewer's suggestion, we have now carried out co-IPs to show full-length JARID2 interacts with EZH2 but Δ N-JARID2 does not (Fig S3 B,C in the revised MS).

However, we would also like to reiterate that multiple papers do show that the C-terminal of JARID2 binds with very low affinity (or not at all) to EZH2 (A summary of these data is given in following table).

Publication	Species	PRC2 subunit tested	N-term	C-term
Cooper et al. (Data provided in supplementary) 2016	Mouse	EZH2	+++	-
Son et al.	Mouse	EZH2	+++	+
deRoche et al. (2014)	Mouse	EZH2	+++	-
Peng et al.* (2009)	Xenopus	Suz12	-	+++

Regarding the work of Cooper et al. whilst they do cite Pasini et al.'s work, they also carried out this experiment themselves (see Supplementary Figure 6 in Cooper et al. Nature Communications 2016). To the best of our knowledge, it is only Peng et al. (2009) who suggest that the C-term of JARID2 interacts with polycomb. Peng et al also suggest that the N-terminus has no interaction with polycomb, but there are multiple other papers that do report this interaction. Possibly, this discrepancy can be explained by use of different model species (Xenopus) used by Peng et al and the use of Suz12, instead of EZH2 antibody.

2b: The authors infer that genes affected by loss of JARID2 in human keratinocytes are PRC2 targets based on distribution of PRC2 in mouse ESC cells. This should be confirmed in the relevant cell type or at least in the relevant species.

We are slightly confused by the reviewer's comments regarding showing enrichment of PRC2 in the same species. The PRC2 enrichment shown in (Previous Fig 5D and revised Fig 4D) is from human keratinocytes and human ES cells. Taking reviewer's suggestion on board, the JARID2 enrichment data is now replaced with human JARID2 ChIP-seq data (Fig 5C in original manuscript and Fig 4C in the revised version).

2c: If the authors wish to make the point that downregulated genes are disproportionately affected they should show in Fig. 5C and D the profile for the subset of upregulated genes not just the randomly selected control.

We agree with reviewer's suggestion and we have now replaced the random control with the up-regulated genes. This was a helpful suggestion and further supports our conclusion that down-regulated genes are preferential targets of polycomb proteins.

2c: In Pg. 13 the authors argue that lack of global changes in H3K27me3 in KOs demonstrate that PRC2 targeting and activity is not affected by the mutation. While global changes in activity would be detectable (although with low sensitivity) in a WB, altered targeting would not.

This is a valid point and we have now edited the text to include this possibility (Page 14, paragraph 3, line 24-25; page 15, paragraph 1, lines 1-5, lines 13-17).

3) Transcriptional effects of Δ N-JARID2

3a) The authors suggest that all changes in gene expression in this model are due to loss of the truncated Δ N-JARID2 version but FL-JARID2 is also present in these cells and could contribute to the phenotype.

The FL-JARID2 present in these cells is at very low or undetectable level. But to rule out the possibility that the effects seen are due to low levels of FL-JARID2, we have now exogenously expressed FL-JARID2 in the JARID2 KO cells. We do not see the same effect as Δ N-JARID2, supporting our hypothesis that the differentiation phenotype in JARID2 KO is primarily a function of Δ N-JARID2. This data has been included in the text (page 12, paragraph 2; page 13, paragraph 1) and in the revised fig 3D and E.

3b) The conclusion that Δ N-JARID2 is a transcriptional activator that acts via AP-1 is supported very indirectly by the data and would require much additional work to prove. I suggest that the authors tone down these conclusions or provide the additional data.

We have amended the text to take into account the reviewers comments. In particular, we have now toned down our conclusions, cited a previous paper and added additional co-immunoprecipitation data for Δ N-JARID2 and AP-1 protein c-Jun (Page 18, paragraph 2; page 19, paragraph 1; page 16, paragraph 2; page 17, paragraph 1; page 20, Appendix fig S4).

Minor points

- Fig. 1C: This blot should include the full-length JARID2 band to show effectiveness of the siRNA on the canonical isoform. What are the units for "relative expression"?

We have now shown the 140kDa region of the western blot. In HaCaT cells we could detect only a very low or undetectable level of 140kDa band. In addition, Fig 1C (Fig 1D in the revised manuscript) is a transient knockdown experiment and to achieve high transfection efficiency, these experiments were done on low number of cells. Therefore we cannot see 140kDa band in these experiments. However, we have repeated these experiments in two other lines HEK293T (Fig 1D) and K562 (Fig EV1B) where the 140kDa band is visible and siRNA efficiency is apparent in the knockdown of the 140kDa band (Fig 1D, Fig EV1B).

The relative expression is measured w.r.t to GAPDH levels. This information is included in the new Fig EV1A.

- Fig. 2B: To support this conclusion it should be shown that the KO mutants still express same levels of full length JARID2 mRNA.

This data has been included in revised Figure EV2C. As shown in the figure, the KO mutant expresses the same level of full-length JARID2 mRNA.

- Pg. 9: The statement "in a blot with anti-Flag antibody, which will detect the N-terminal tag on JARID2, we could not detect the LMW JARID2 form" should be supported by data

This has been included in the revised manuscript (Figure EV2G).

2nd Editorial Decision

17th July 2018

Thank you for submitting a revised version of your manuscript to The EMBO Journal. It has now been seen by all three original referees and their comments are shown below.

As you will see they all find that the major criticisms have been sufficiently addressed and recommend the manuscript for publication (pending clarification of few minor issues from ref #3). I would therefore invite you to submit a final revised version in which you address these remaining points as well as the following editorial issues concerning text and figures

REFeree REPORTS

Referee #1:

In the revised manuscript, entitled "A novel form of JARID2 is required for differentiation in Lineage-Committed Cells", the authors addressed most of my comments satisfactorily. For example, they validated the existence of the truncated form of Jarid2 by multiple means, providing evidence that it is a cleavage product of full length Jarid2 rather than alternative splicing or isoform. As mentioned in my original review, it will be very interesting to further characterise this novel form of Jarid2 in CHIP binding analyses. However, the authors point to the fact that these future studies may prove to be technically challenging.

Referee #2:

This study reports a novel low-molecular weight form of Jarid2, and highlights its role in differentiation of lineage committed cells. This is an important and novel finding, as Jarid2 has previously been thought to be important in early development, but down regulated upon ES cell differentiation.

Major Issues

1. The new manuscript has used additional antibodies, alternative siRNAs to confirm the identity of the of the 80kDa band as C-terminal Jarid2. The western blot using the alternate C-terminal antibody (GTX129020) has not come out very clearly in figure EV1-B, however the complimentary use of a mass spectrometry approach to validate the identity of the 80kDa band as Δ N-Jarid2 band significantly strengthens this finding.
2. The authors highlight that Jarid2 isoform 1 mRNAs are detectable in lineage committed cells, although at lower levels than observed in stem cells. Furthermore they suggest a plausible explanation for the dichotomy, with Δ N-Jarid2 having increased stability, possibly as a result of losing N-terminal ubiquitination sites, but recognise that further study is needed.
3. The authors have revised the introduction to reflect this feedback.
4. The authors have cited a relevant publication, and have modified their model to reflect the results of immunoprecipitation experiments.

Overall the manuscript has been considerably improved, with substantial additional experimental data included. Specifically, the use of the complimentary mass spectrometry approach to confirm identity of the 80kDa as C-terminal Jarid2 is important and significantly increases the strength of this manuscript. The authors have also satisfactorily addressed the other major and minor issues.

Referee #3:

I am overall mostly satisfied with the improvements made in this revised manuscript and I do not have major objections to its publication. I do note a few minor points that I think the authors might want to consider as they prepare a final version.

- Fig. S3B: why is the input band of different size than the pull-down band for the 140 kDa JARID2?
- The authors might want to cite Chen & Liu Mol Cell 2018, who show the structure of an N-terminal region of JARID2 associated with core PRC2 and validate the interaction.
- Fig. 1C: would be good to know what is the background in a negative control. The authors could include qRT-PCR from KO cells.
- Fig. EV2F-G: I don't understand this figure. Isn't the band at 140 kDa in EV2F FL-JARID2? Shouldn't that react with the FLAG antibody in EV2G?
- Fig. 4D: this should indicate that the plot is for EZH2, for clarity
- Fig. 4D: the enrichment for EZH2 in the human keratinocytes is marginal

Reviewer 3:**1. Fig. S3B: why is the input band of different size than the pull-down band for the 140 kDa JARID2?**

In this experiment, JARID2 was immunoprecipitated from cells transfected with an empty vector and a vector expressing FLAG-tagged full-length JARID2 (2nd and 3rd lanes). These immunoprecipitated samples, along with input (from un-transfected cells, 1st lane), were blotted with anti-FLAG antibody to specifically detect exogenously expressed JARID2. We suspect that, because of the lack of FLAG-tag in the un-transfected cells, anti-FLAG antibody cross-reacts and displays a non-specific band in the input lane.

2. The authors might want to cite Chen & Liu Mol Cell 2018, who show the structure of an N-terminal region of JARID2 associated with core PRC2 and validate the interaction.

Taking the reviewer's comments into account, we have revised the manuscript to include the suggested reference (page 15 and page 17) as an additional supporting evidence to show that N-terminal region of JARID2 is associated with PRC2.

3. Fig. 1C: would be good to know what is the background in a negative control. The authors could include qRT-PCR from KO cells.

In CRISPR KOs, mRNA levels do not change. In fact, in our previous revision, as per reviewer's suggestion, we had included a figure (EV2C) to show similar mRNA levels in WT and KO cells. As a result qRT-PCR from KO cells will not serve as a good negative control. Alternative solution would have been to carry out a qRT-PCR on a cell type that does not express JARID2. However, due to ubiquitous nature of JARID2 it is difficult to find such a negative control.

4. Fig. EV2F-G: I don't understand this figure. Isn't the band at 140 kDa in EV2F FL-JARID2? Shouldn't that react with the FLAG antibody in EV2G?

EV2F and EV2G experiments are carried out in two different cell lines. Experiment in EV2F is from HEK293T cells where FLAG-tag vector expression is at much higher level than in HaCaT cells (EV2G). As a result in EV2F, in spite of JARID2 cleavage, we can still detect 140kDa band with anti-FLAG antibody.

5. Fig. 4D: this should indicate that the plot is for EZH2, for clarity**Fig. 4D: the enrichment for EZH2 in the human keratinocytes is marginal.**

Taking reviewer's comment into account, we have indicated using Y-axis label that the plot in 4D is that of EZH2. Lower level of EZH2 enrichment in human keratinocytes as compared to ES cells might be due to lower expression of polycomb proteins in these cells. We will like to note that we see similar enrichment when we use randomly selected genes as a control (Figure in the first version of manuscript).

Thank you for submitting the final version of your manuscript, I'm pleased to inform you that your study has been accepted for publication in The EMBO Journal.

Corresponding Author Name: Aditi Kanhere

Manuscript Number: EMBOJ-2017-98449